# High-moisture alfalfa silage fermentation: a comparative study on the impact of additives including formic acid, *Lactobacillus plantarum*, cinnamon essential oil, and wood vinegar

Xingquan Yan,[1,2,3] Jian Bao,[4] Muqier Zhao,[1,2,3] Zhuotong Liu,[5] Mingjiu Wang,[6] Jingyi Liu,[1,2,3] Pengbo Sun,[1,2,3] Yushan Jia,[1,2,3] Gentu Ge,[1,2,3] Zhijun Wang[1,2,3]

**ABSTRACT**  Applying additives to high-moisture alfalfa silage is crucial for ensuring fermentation quality. The study selected two conventional silage additives (formic acid [FA] and *Lactobacillus plantarum* [LP]) and two additives with the potential to improve silage fermentation quality (cinnamon essential oil [CEO] and wood vinegar [WV]) as the subjects of the research, with the control group sprayed with an equal amount of distilled water (CK). We compared the fermentation products of high-moisture alfalfa silage at 3 d, 7 d, 15 d, 30 d, and 60 d with different additives and evaluated the dry matter (DM) *in vitro* digestibility, crude protein (CP) *in vitro* digestibility, and the bacterial community composition, co-occurrence network, function, and phenotype prediction of the alfalfa silage at 60 d of fermentation. The results indicate that after the application of additives, the pH of the alfalfa silage decreased more rapidly, and it retained more DM and CP, produced more lactic acid (LA), and reduced the levels of acid detergent fiber, neutral detergent fiber, acetic acid, and butyric acid, thereby enhancing *in vitro* DM digestibility and *in vitro* CP digestibility. In terms of bacterial community composition, LA bacteria are virtually absent in the CK group, with harmful microorganisms such as *Pantoea* predominating. LP, CEO, and WV are enriched with a large number of bacteria beneficial to LA fermentation, such as *Lactobacillus*, *Pediococcus*, *Lactococcus*, and *Weissella*. At the same time, the bacterial co-occurrence network structure tends to simplify, and it significantly reduces Potentially Pathogenic and Mobile Element Containing. In summary, LP, CEO, and WV are more effective additives for high-moisture alfalfa silage, capable of effectively improving the quality and safety of the silage.

**IMPORTANCE**  During the ensiling process of alfalfa, the growth and proliferation of harmful microorganisms can lead to a decline in silage quality. In this study, we report for the first time the improvement effect of cinnamon essential oil (CEO) and wood vinegar (WV) on the quality of alfalfa silage, which outperforms formic acid. This provides a reference for the development of novel silage additives. The results indicate that the addition of *Lactobacillus plantarum*, CEO, and WV can enhance the abundance of beneficial microorganisms, including *Lactobacillus*, *Pediococcus*, *Lactococcus*, and *Weissella*, while inhibiting the growth of harmful microorganisms such as *Pantoea*, thereby effectively improving the quality and safety of alfalfa silage.

**KEYWORDS**  alfalfa, silage, bacterial community, cinnamon essential oil, wood vinegar

Alfalfa (*Medicago sativa* L.), known for its high protein content, excellent nutritional quality, and good palatability, is a leguminous forage widely cultivated across the globe. With the rapid development of the livestock industry in China, the demand for high-quality alfalfa continues to grow. Silage, as one of the primary methods of

**Peer Reviewer** Bo Yao, Gansu Agricultural University, Lanzhou, Gansu, China

Address correspondence to Gentu Ge, gegentu@163.com, or Zhijun Wang, zhijunwang321@imau.edu.cn.

The authors declare no conflict of interest.

See the funding table on p. 14.

processing alfalfa, offers the advantages of long-term preservation and being less susceptible to weather fluctuations. However, due to alfalfa's characteristic of having a low content of water-soluble carbohydrates and a high buffering capacity, it results in difficulty in reducing the pH during the fermentation process, which in turn promotes the growth and proliferation of spoilage bacteria (1, 2). Therefore, inhibiting the growth and proliferation of harmful microorganisms during the silage fermentation process is key to preventing the decline in the quality of silage feed.

Plant essential oils, which are secondary metabolites present in various parts of plants such as flowers, stems, leaves, and seeds, have been demonstrated to have an inhibitory effect on the activity of a variety of microorganisms (3). Susanto et al. through a meta-analysis of 17 relevant studies found that as the proportion of plant essential oils added increased, the fermentation quality of silage feed was significantly improved, and the number of molds in the silage feed was also markedly reduced (4). Roberto et al. also discovered that the addition of plant essential oils to sugarcane silage can effectively reduce the quantity of molds and yeasts and enhance the aerobic stability of the silage feed (5). Research results by Li et al. also indicate that *Amomum villosum* essential oil possesses the potential to enhance the fermentation quality of silage feed (6). Additionally, plant essential oils have also been demonstrated to enhance the activity of rumen microorganisms in livestock, improve the efficiency of nutrient utilization, and reduce methane production in dairy cows. At the same time, they can improve the antioxidant and anti-inflammatory capabilities in weaned piglets, thereby enhancing their stress resistance (7). Cinnamon essential oil (CEO), which is derived from cinnamon, contains compounds such as cinnamaldehyde, as well as polysaccharides, polyphenols, and flavonoids (8). Research has indicated that cinnamon essential oil can inhibit the growth rate of *Aspergillus flavus* and *Fusarium moniliforme* in rice and effectively prevent the production of aflatoxin B1 and fumonisin B1 (9). Previous research has found that the addition of cinnamon essential oil to ensiled peas can reduce the content of acetic acid (AA) and ammonium nitrogen in the silage feed, increase the content of crude protein (CP) and dry matter (DM), and effectively suppress the growth of molds during aerobic exposure (10). However, there have been no studies reported to date on the effects of plant essential oils, especially cinnamon essential oil, in high-moisture alfalfa silage.

Wood vinegar (WV) is a brown liquid produced from the pyrolysis of biomass materials, containing over 200 different compounds, including furans, organic acids, esters, phenols, alcohols, and pyran derivatives (11). Among the components of wood vinegar, organic acids and phenolic compounds are the main constituents, exhibiting significant antimicrobial properties against common bacteria found in silage feeds, such as *Escherichia*, *Staphylococcus*, and *Pseudomonas* (12–14). Wu et al. discovered that the addition of 1%–2% wood vinegar can reduce the quantity of *Escherichia coli* and molds, decrease bacterial diversity, and improve the fermentation quality of sugarcane silage (15). De Souza Araujo et al. also found that wood vinegar can effectively inhibit the growth of yeast at the top of sugarcane (16).

Given that numerous researchers have demonstrated the prominent antimicrobial characteristics of plant essential oils and wood vinegar, we hypothesize that these additives may modify the bacterial community during the fermentation process of silage feed, thereby improving the fermentation quality of high-moisture alfalfa silage. However, to date, there have been no reports on the effects of plant essential oils and wood vinegar on the fermentation quality of high-moisture alfalfa silage. Simultaneously, additives such as formic acid (FA) and lactic acid bacteria (LAB) have also been repeatedly proven to effectively improve the fermentation quality of silage feed (17, 18). Based on this, the present study aims to clarify the effects of cinnamon essential oil, wood vinegar, formic acid, and *Lactobacillus plantarum* (LP) on the fermentation quality and bacterial community of alfalfa silage. The focus is on whether cinnamon essential oil and wood vinegar, as novel additives, can improve the quality of silage, providing technical support for the development of new types of silage additives.

## MATERIALS AND METHODS

### Ensiling processes

On 16 August 2023, alfalfa was harvested from the Hailutu Experimental Station of Inner Mongolia Agricultural University (Hohhot, China), with the alfalfa at the bud stage at the time of cutting, and a stubble height of 5 cm was maintained. The alfalfa was chopped to a length of 1–2 cm and then wilted until it reached a moisture content of 75% before ensilage. Based on fresh matter, 5.00 kg of wilted alfalfa was treated with the following solutions: 100 mL of distilled water (CK), a mixture of 25 mL of FA plus 75 mL of distilled water, 0.025 g of LP ($1 \times 10^6$ cfu/g) plus 100 mL of distilled water, a mixture of 10 mL of CEO plus 90 mL of distilled water, and 100 mL of WV. After thoroughly mixing the additives with the alfalfa, the mixture was packed into polyethylene bags ($30 \times 40$ cm), with approximately 300 g in each bag, which were then vacuum sealed and stored at room temperature. For each treatment, 15 silage samples were prepared, and for the determination of fermentation parameters, three random samples from each treatment were selected on days 3, 7, 15, 30, and 60 of fermentation. Additionally, the bacterial community of the alfalfa silage samples after 60 d of fermentation was also analyzed.

### Fermentation characteristics and *in vitro* fermentation parameter analysis

Twenty gram of silage material was randomly selected and soaked in 180 mL of 0.9% sterile saline for 15 min, then diluted to the order of $10^{-1}$ to $10^{-6}$ in an ultra-clean bench. The number of LAB was enumerated on de Man, Rogosa, and Sharpe agar after incubation at 30°C for 48 h under anaerobic conditions (LRH-250, Shanghai, China). Coliform bacteria were counted on Violet Red Bile Agar incubated at 30°C for 48 h. Yeasts and molds were counted on Rose Bengal Agar after being incubated at 28°C for 48 h under aerobic conditions.

The chemical properties of alfalfa before and after ensilage were analyzed. Ten gram of alfalfa silage was mixed with 90 mm of distilled water, and after blending for 2 min using a stomacher homogenizer, the mixture was filtered. The pH of the filtrate was measured using a glass electrode pH meter. The concentrations of lactic acid (LA), AA, propionic acid (PA), and butyric acid (BA) were determined using high-performance liquid chromatography (19). The samples were dried at 65°C for 48 hours for the determination of DM content. The CP content was determined using an Automatic Kjeldahl Nitrogen Analyzer (GK-500, GLKRI, China) (20). The contents of neutral detergent fiber (NDF) and acid detergent fiber (ADF) were measured using an Ankom A2000i Fiber Analyzer (A2000i, Ankom Technology, Macedon, NY, United States) (21).

Referencing the method of Yi et al. (22), the alfalfa samples after 60 d of ensilage were evaluated using a constant-temperature artificial rumen incubator (BZ-SHH-W21, Biaozhuo Scientific Instruments Co., Ltd., Shanghai, China). After incubating the ensiled alfalfa samples at 39°C for 72 hours, they were removed, the filter bags were thoroughly rinsed with water, and then the samples were dried at 65°C for 48 hours. Then, the CP content of the digested alfalfa samples was measured using the Kjeldahl nitrogen method, and the *in vitro* dry matter digestibility (IVDMD) and *in vitro* crude protein digestibility (IVCPD) were calculated based on the weights and CP contents of the samples before and after digestion. Each sample was measured in triplicate, nine times in total. The formulas for IVDMD and IVCPD are as follows:

$$IVDMD = ((W1 - W2) / W1) \times 100\%.$$

W1 is the dry weight of the sample before digestion, and W2 is the dry weight of the sample after digestion.

$$IVCPD = ((P1 - P2) / P1) \times 100\%.$$

P1 is the CP content of the sample before digestion, and P2 is the CP content of the sample after digestion.

## DNA extraction and high-throughput sequencing

DNA was extracted from alfalfa silage samples according to the kit instructions (D4015, Omega Inc., Norcross, GA, United States), and the quality of the extraction was checked by NanoDrop 2000 luminometer. The extracted DNA was amplified by PCR technique. The primers 799F (5′-AACMGGATTAGATAC CCKG-3′) and 1193R (5′- ACGTCATCCCCACCT TCC-3′) were used to amplify the bacterial V3–V4 region, followed by a PCR. PCR was conducted as follows: 95℃ for 3 min, 27 cycles (95℃, 30 s; 55℃, 30 s; 72℃, 45 s), then an extension at 72℃ for 10 min, ending at 4℃. Subsequently, the PCR products were extracted from a 2% agarose gel and purified. Finally, purified amplicons were used for sequencing on the Illumina MiSeq 2 × 300 platform by Majorbio Bio-Pharm Technology Co. Ltd. (Shanghai, China).

## Data analysis

The chemical properties, fermentation quality, and *in vitro* digestibility of alfalfa silage were analyzed using one-way analysis of variance (ANOVA; SPSS Statistics 26). The parameters were tested for normality (Shapiro-Wilk test) and homogeneity of variance. The differences among the treatment groups were assessed by Duncan's multiple range test and the Kruskal-Wallis *H* test. The graph was created using Origin 2021.

The sequencing data were rarefed to compare all samples at the same sequencing depths. The bacterial sequences were analyzed using Majorbio Bio-Pharm Technology Co. Ltd.'s cloud platform (https://cloud.majorbio.com/page/tools.html). Alpha diversity indices were calculated, beta diversity was assessed by calculating the Bray-Curtis distance matrix, and differences in microbial community composition between treatment groups were observed using principal coordinates analysis (PCA). Co-occurrence networks were constructed at the genus level for bacteria with a relative abundance of at least 0.01%. The similarity matrix was created by employing the Pearson correlation coefficient. A network was constructed using an equivalent random matrix theory threshold of 0.50, with a *P*-value less than 0.05. Topological properties were analyzed using the MENAP (http://ieg4.rccc.ou.edu/mena/). Visualized networks using Gephi (0.10).

## RESULTS AND DISCUSSION

### Chemical properties of the silage material

The results of chemical characterization and microbial counts of alfalfa before silage are shown in Table 1. The DM and CP contents of alfalfa before silage were 25.49% and 19.22%, respectively. The ADF and NDF contents were higher at 30.12% and 51.65%, respectively. The water-soluble carbohydrates (WSC) content was low, at 2.49%, which is below the theoretical requirement of 6%–7% (23). At the same time, the number of LAB is only 4.89 log10 cfu/g, which is lower than the number of lactic acid bacteria required for quality silage (>5.00 log10 cfu/g) (24). Yeasts, molds, and coliform bacteria reached 5.23, 3.66, and 5.01 log10 cfu/g, respectively.

### Fermentation characteristics and *in vitro* digestibility of alfalfa silage

The fermentation characteristics of high-moisture alfalfa silage in this study are presented as shown in Table 2. The type of additive, storage time, and the interaction between the two had significant effects on the contents of CP, NDF, ADF, LA, AA, and PA in alfalfa silage, as well as on the pH and the LA/AA ratio (*P* < 0.05). The contents of DM and CP were consistently the lowest in the control group (CK) across all storage times. Under high-moisture conditions, ensiling can lead to the proliferation of harmful microorganisms such as *Clostridium* species, which compete with lactic acid bacteria

**TABLE 1** Chemical characteristics and microbiological counts of alfalfa before silage[a]

| Items | Alfalfa |
|---|---|
| DM (%FM) | 25.49 ± 1.38 |
| CP (%DM) | 19.22 ± 0.84 |
| WSC (%DM) | 2.49 ± 0.77 |
| Ash (%DM) | 9.53 ± 0.49 |
| NDF (%DM) | 51.65 ± 0.38 |
| ADF (%DM) | 30.12 ± 0.28 |
| LAB (log10 cfu/g FM) | 4.89 ± 0.77 |
| Yeasts (log10 cfu/g FM) | 5.23 ± 0.94 |
| Molds (log10 cfu/g FM) | 3.66 ± 0.29 |
| Coliform bacteria (log10 cfu/g FM) | 5.01 ± 1.06 |

[a]FM, fresh material.

and can break down and utilize the proteins in forage, resulting in the production of biogenic amines and ammonia (25). Simultaneously, because ammonium nitrogen is a basic substance, the substantial accumulation of ammonium nitrogen during the later stages of fermentation is likely the cause of the significant increase in pH value for CK starting at 30 d, and even reaching a pH of 7.21 at 60 d, which was significantly higher than that of the other treatment groups ($P < 0.05$). The pH of FA treatment decreased the fastest, and throughout the entire storage period, the pH of FA was the lowest. The pH of CEO initially decreased more slowly but began to drop rapidly by 30 d, and by 60 d, it was similar to the pH of the LP treatment group. Li et al. also found that the addition of *Amomum villosum* essential oil to paper mulberry silage significantly reduced the pH, and its effect was even superior to that of lactic acid bacteria additives (6). The pH value is a direct indicator of the extent of fermentation in silage feed. For acceptable legume silage feed, the pH value typically falls within the range of 4.3–5.00 (26). In this study, after 60 d of fermentation, only the LP, CEO, and FA treatment groups had pH values within the 4.3–5.00 range. After 60 d of storage, the NDF content in the FA treatment group was significantly higher than in the other treatment groups ($P < 0.05$), while the ADF content was the highest in the CK group ($P < 0.05$).

Lactic acid, which is produced through the metabolism of lactic acid bacteria, is the primary factor for the reduction in pH of silage feed. The LA content in the FA, LP, CEO, and WV treatments all began to rise rapidly after 30 d of storage. By day 60, CEO and LP were the treatments with the highest LA content, reaching 6.92% and 6.07%. This indicates that the application of additives is beneficial to lactic acid fermentation, with the addition of plant essential oils and LP showing the most pronounced effects, which is similar to the findings of previous research (6, 27). An appropriate amount of AA can effectively lower the silage feed pH and enhance its aerobic stability. However, when AA content is too high, it can negatively affect the palatability and fermentation quality of the silage feed (28). In this study, the AA content in CK was the highest in all periods except for 15 d, and after 60 d of fermentation, CK had the highest AA content and the lowest LA/AA ratio, at 11.96% and 0.05, respectively, which were significantly different from the other treatment groups ($P < 0.05$). In silage feed with good fermentation quality, the content of PA and BA should be as low as possible or ideally non-existent (29). In this study, BA was only detected in CK at 60 d of ensiling, with a content of 0.66%. PA content was relatively low initially but gradually increased with the extension of fermentation time. By day 60, PA was detected in all treatment groups, with the highest content in CK, reaching 0.64%, which was significantly different from the other treatment groups ($P < 0.05$).

Then, we compared the IVDMD and IVCPD of the different treatment groups using the *in vitro* digestion method (Fig. 1). After the application of additives, the IVDMD and IVCPD of alfalfa silage both underwent significant changes. The CK treatment group had the lowest IVDMD, only reaching 40.16%, which was significantly different from the groups treated with additives ($P < 0.05$). In this study, as the contents of NDF and ADF decreased,

**TABLE 2** Fermentation characteristics of alfalfa silage[a,c]

| Items | Treatments | Store time | | | | | P value | | |
|---|---|---|---|---|---|---|---|---|---|
| | | 3 d | 7 d | 15 d | 30 d | 60 d | G | D | G*D |
| DM | CK | 25.45 ± 1.52A | 23.78 ± 1.35C | 24.4 ± 1.11B | 22.09 ± 1.24C | 22.33 ± 0.85C | <0.001 | <0.001 | 0.171 |
| (%FM) | FA | 26.81 ± 2.78A | 25.74 ± 0.89B | 26.38 ± 0.69A | 24.82 ± 0.96AB | 24.2 ± 0.3.00B | | | |
| | LP | 26.74 ± 0.88A | 28.36 ± 1.22A | 26.17 ± 0.82A | 26.53 ± 1.78A | 24.00 ± 0.65B | | | |
| | CEO | 26.94 ± 1.95A | 26.41 ± 0.67B | 27.53 ± 0.56A | 27.99 ± 3.05A | 25.62 ± 0.18A | | | |
| | WV | 27.14 ± 0.77A | 25.31 ± 0.73BC | 26.68 ± 0.53A | 24.8 ± 0.79AB | 24.31 ± 0.19B | | | |
| CP | CK | 17.35 ± 0.51C | 18.40 ± 0.44C | 16.88 ± 0.20B | 16.53 ± 0.19C | 16.98 ± 0.72C | <0.001 | <0.001 | 0.011 |
| (%DM) | FA | 19.47 ± 0.56AB | 19.48 ± 0.19AB | 18.16 ± 0.45A | 17.66 ± 0.12B | 17.80 ± 0.56BC | | | |
| | LP | 18.35 ± 0.98BC | 18.74 ± 0.73BC | 18.61 ± 0.81A | 17.12 ± 0.41BC | 19.07 ± 0.50A | | | |
| | CEO | 20.23 ± 0.68A | 19.4 ± 0.52AB | 18.81 ± 0.03A | 17.74 ± 0.28B | 18.37 ± 0.64AB | | | |
| | WV | 19.89 ± 1.53AB | 19.65 ± 0.15A | 18.68 ± 0.8A | 19.41 ± 0.61A | 18.66 ± 0.66AB | | | |
| NDF | CK | 48.84 ± 0.59A | 48.45 ± 0.81AB | 45.14 ± 0.59A | 43.74 ± 0.35AB | 43.48 ± 0.45B | <0.001 | <0.001 | <0.001 |
| (%DM) | FA | 46.55 ± 0.34B | 48.56 ± 0.68A | 44.95 ± 0.38A | 45.01 ± 1.06A | 47.5 ± 0.81A | | | |
| | LP | 48.72 ± 0.31A | 43.95 ± 0.45C | 43.80 ± 0.41B | 39.10 ± 1.36C | 37.48 ± 0.99C | | | |
| | CEO | 45.95 ± 0.22B | 47.41 ± 0.38B | 42.76 ± 0.22C | 42.18 ± 0.65B | 38.55 ± 0.90C | | | |
| | WV | 44.17 ± 0.43C | 44.85 ± 0.57C | 40.07 ± 0.81D | 39.84 ± 0.42C | 37.20 ± 0.60C | | | |
| ADF | CK | 27.97 ± 0.37B | 28.09 ± 0.59A | 26.59 ± 0.40A | 27.34 ± 0.2A | 28.36 ± 0.32A | <0.001 | <0.001 | <0.001 |
| (%DM) | FA | 25.93 ± 0.30D | 27.98 ± 0.46A | 25.34 ± 0.44BC | 29.18 ± 6.32A | 26.64 ± 0.74B | | | |
| | LP | 29.07 ± 0.45A | 28.73 ± 0.61AB | 26.01 ± 0.48AB | 25.7 ± 1.18AB | 24.00 ± 1.10C | | | |
| | CEO | 26.78 ± 0.25 CD | 28.11 ± 0.22A | 24.86 ± 0.21C | 24.84 ± 0.4AB | 24.10 ± 1.13C | | | |
| | WV | 26.95 ± 0.80C | 29.12 ± 0.46A | 25.20 ± 0.64BC | 20.57 ± 0.41B | 19.89 ± 0.57D | | | |
| pH | CK | 6.65 ± 0.07A | 6.63 ± 0.15A | 6.57 ± 0.19A | 6.98 ± 0.23A | 7.21 ± 0.12A | <0.001 | <0.001 | <0.001 |
| | FA | 5.44 ± 0.13D | 4.90 ± 0.05D | 4.72 ± 0.12E | 4.38 ± 0.09C | 4.47 ± 0.08C | | | |
| | LP | 5.99 ± 0.11C | 5.60 ± 0.19C | 5.34 ± 0.01D | 5.29 ± 0.01BC | 4.97 ± 0.21B | | | |
| | CEO | 6.22 ± 0.08B | 6.05 ± 0.07B | 5.92 ± 0.14B | 5.51 ± 0.01AB | 4.95 ± 0.09B | | | |
| | WV | 6.00 ± 0.05C | 5.95 ± 0.04B | 5.58 ± 0.03B | 5.46 ± 0.07ABC | 5.12 ± 0.21B | | | |
| LA | CK | 0.92 ± 0.05D | 2.44 ± 0.04A | 2.00 ± 0.04C | 2.00 ± 0.05B | 0.57 ± 0.01D | <0.001 | <0.001 | <0.001 |
| (%FM) | FA | 1.15 ± 0.09C | 2.32 ± 0.05B | 2.28 ± 0.12B | 1.21 ± 0.06C | 4.62 ± 0.04C | | | |
| | LP | 0.99 ± 0.03D | 1.53 ± 0.08C | 2.81 ± 0.01A | 2.85 ± 0.03A | 6.07 ± 0.79B | | | |
| | CEO | 1.64 ± 0.06B | 1.44 ± 0.02D | 2.32 ± 0.04B | 2.4 ± 0.05ABC | 6.92 ± 0.09A | | | |
| | WV | 3.10 ± 0.01A | 0.38 ± 0.02E | 1.63 ± 0.05D | 2.53 ± 0.06AB | 4.67 ± 0.02B | | | |
| AA | CK | 2.48 ± 0.1A | 5.00 ± 0.25A | 4.41 ± 0.11A | 4.34 ± 0.12A | 11.96 ± 0.20A | <0.001 | <0.001 | <0.001 |
| (%FM) | FA | 1.60 ± 0.02C | 3.85 ± 0.27B | 2.47 ± 0.27C | 3.10 ± 0.66B | 3.20 ± 0.19B | | | |
| | LP | 1.92 ± 0.24B | 2.56 ± 0.07C | 4.70 ± 0.1A | 4.69 ± 0.05A | 4.28 ± 1.34B | | | |
| | CEO | 1.76 ± 0.10BC | 1.81 ± 0.07D | 2.64 ± 0.08AB | 3.37 ± 0.22B | 3.39 ± 0.09B | | | |
| | WV | 0.35 ± 0.01D | 0.54 ± 0.17E | 1.38 ± 0.07C | 1.52 ± 0.08C | 3.99 ± 0.05B | | | |
| LA/AA | CK | 0.37 ± 0.02C | 0.49 ± 0.02B | 0.45 ± 0.01D | 0.46 ± 0.02BC | 0.05 ± 0.00D | <0.001 | <0.001 | <0.001 |
| | FA | 0.71 ± 0.05AB | 0.61 ± 0.03AB | 0.93 ± 0.05B | 0.40 ± 0.06C | 1.44 ± 0.08B | | | |
| | LP | 0.52 ± 0.08BC | 0.60 ± 0.03AB | 0.60 ± 0.01C | 0.61 ± 0.01ABC | 1.47 ± 0.24B | | | |
| | CEO | 0.93 ± 0.05AB | 0.80 ± 0.03A | 0.88 ± 0.02B | 0.71 ± 0.04AB | 2.05 ± 0.03A | | | |
| | WV | 8.82 ± 0.35A | 0.75 ± 0.24A | 1.18 ± 0.03A | 1.67 ± 0.11A | 1.17 ± 0.01C | | | |
| PA | CK | 0.05 ± 0.00 | 0.09 ± 0.01B | 0.06 ± 0.00C | 0.05 ± 0.01B | 0.64 ± 0.08A | <0.001 | <0.001 | <0.001 |
| (%FM) | FA | ND | 0.16 ± 0.02A | 0.11 ± 0.01A | 0.22 ± 0.01A | 0.34 ± 0.01C | | | |
| | LP | ND | ND | 0.05 ± 0.01D | 0.05 ± 0.00B | 0.34 ± 0.02C | | | |
| | CEO | ND | 0.02 ± 0.02C | 0.11 ± 0.00A | 0.10 ± 0.01AB | 0.42 ± 0.01B | | | |
| | WV | ND | 0.01 ± 0.02C | 0.08 ± 0.01B | 0.13 ± 0.01A | 0.36 ± 0.00C | | | |
| BA | CK | ND | ND | ND | ND | 0.66 ± 0.02 | _[b] | _[b] | _[b] |
| (%FM) | FA | ND | ND | ND | ND | ND | | | |
| | LP | ND | ND | ND | ND | ND | | | |
| | CEO | ND | ND | ND | ND | ND | | | |
| | WV | ND | ND | ND | ND | ND | | | |

[a]FM, fresh material. Different capital letters indicate significant differences between treatments ($P < 0.05$).
[b]"−" indicates that variance analysis was not performed on this data.
[c]ND, no detection.

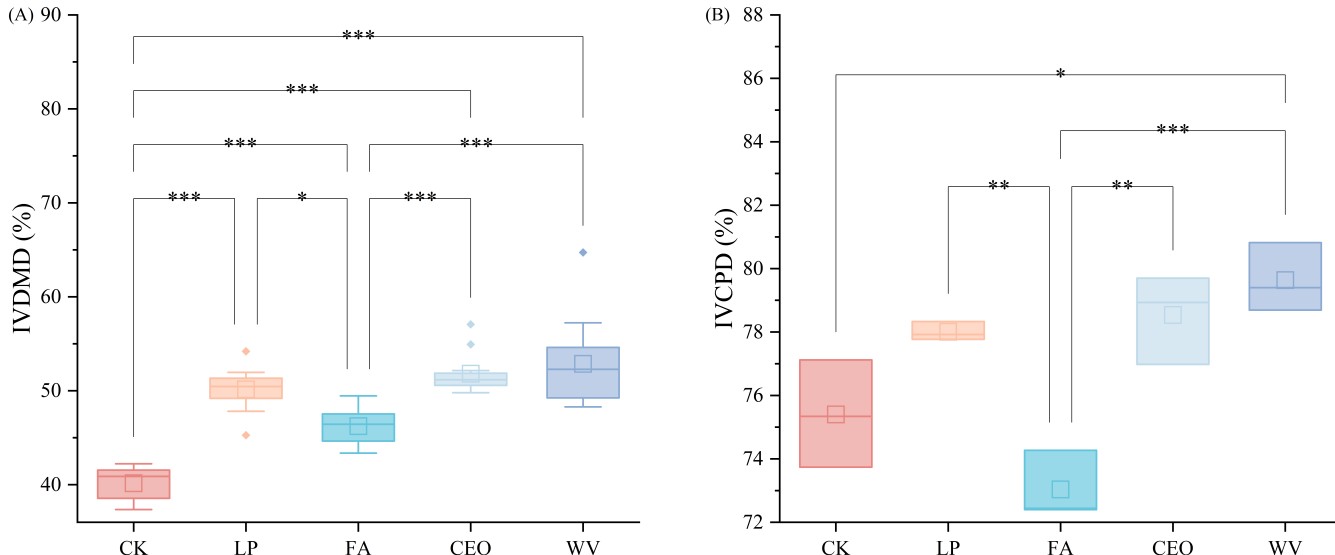

**FIG 1** IVDMD (A) and IVCPD (B) of alfalfa silage after 60 d of fermentation. * *P* < 0.05, ** *P* < 0.01, and *** *P* < 0.001, same below.

the IVDMD of alfalfa silage also increased, similar to the findings of Du et al. (30). Fiber content is a major factor affecting silage feed digestibility, and reducing fiber content significantly increases the proportion of degradable DM (31). WV and CEO were the two treatments with the highest IVDMD, reaching 52.86% and 51.81%, respectively, which were increases of 31.62% and 29.01% over CK. FA's IVCPD was the lowest, at only 73.04%, but the difference from CK was not significant (*P* > 0.05). WV was the treatment group with the highest IVCPD, reaching 79.64%, which was a significant increase of 5.62% over CK (*P* < 0.05). This suggests that the WV treatment can effectively increase the proportion of soluble protein in alfalfa silage. Since the application of additives improved the fermentation quality of alfalfa silage, it suppressed the fermentation of undesirable microorganisms, reduced the loss of DM and CP, and consequently increased IVDMD and IVCPD (22). Additionally, natural plant essential oils have been proven to improve feed utilization efficiency by modulating the activity of rumen microorganisms and altering rumen metabolism (32). It has also been reported that adding wood vinegar to the diet of finishing pigs can increase the total tract nutrient digestibility of dry matter and nitrogen (33). This could also be one of the reasons why the IVDMD and IVCPD are higher in CEO and WV in this study.

## Bacterial diversity in alfalfa silage

To explore the effects of different additive treatments on the bacterial community in alfalfa silage, we tested the bacterial community in alfalfa silage after 60 d of fermentation. With the increase in the number of reads, the dilution curves of the Sobs index and Shannon index approach saturation, indicating that the sequencing data are adequate and reasonable (Fig. 2A and B). Then, we calculated the bacterial α-diversity indices (Fig. 2C). At 60 d of storage, CK and FA were the two treatments with the highest abundance-based coverage estimator (ACE), Chao, Sobs, and Shannon indices, significantly different (*P* < 0.05) from the other three treatment groups. The ACE, Chao, Sobs, and Shannon indices for LP were the lowest among all treatment groups. The Simpson index showed that LP was significantly higher than the other treatment groups (*P* < 0.05), while CK was the lowest. After adding LP, CEO, and WV, the α-diversity of alfalfa silage significantly decreased, characterized by lower ACE, Chao, Sobs, and Shannon indices and a higher Simpson index. This is mainly because, under anaerobic and low pH conditions, a large number of microorganisms are replaced by lactic acid bacteria, leading to a decrease in the observed species indices (6).

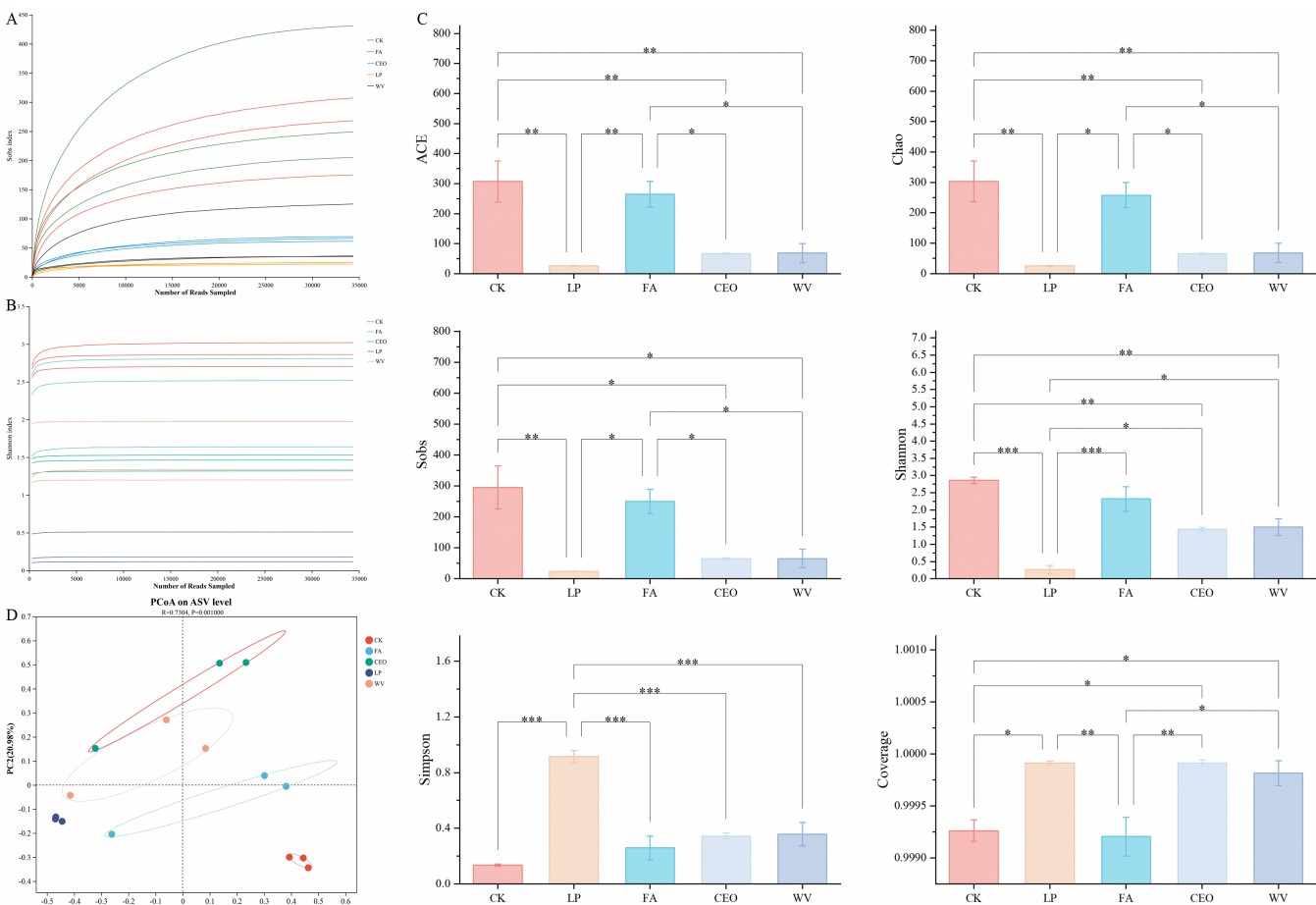

**FIG 2** Rarefaction curves based on observed operational taxonomic units number (A) and Shannon index (B). Alpha-diversity index of bacterial communities (C). Beta diversity of bacterial communities was calculated by PCoA based on the Bray-Curtis distance metric (D). * $P < 0.05$, ** $P < 0.01$, and *** $P < 0.001$.

PCoA was used to compare the differences in bacterial community β-diversity within alfalfa silage (Fig. 2D). CK's markers and those of LP, CEO, and WV are distributed in different quadrants, indicating that the addition of additives has led to a significant change in the β-diversity of the bacterial community in alfalfa silage. During the fermentation process, acid-sensitive aerobes gradually become inactive, which to some extent leads to the separation of bacteria among different treatment groups in PCoA (34).

## Bacterial community composition in alfalfa silage

The composition of the bacterial community in alfalfa silage was analyzed at both the phylum and genus levels, and a one-way ANOVA was conducted on the nine most abundant genera (Fig. 3). At the phylum level, *Firmicutes* and *Proteobacteria* were the two most abundant bacterial groups in all treatment groups, similar to the findings of previous studies (35). The addition of LP, CEO, and WV increased the abundance of *Proteobacteria* phylum bacteria. At the genus level, there were considerable differences in the bacterial community composition among the treatment groups. *Lactobacillus* was the predominant bacteria in the LP treatment, with a relative abundance of 96.16%, which was the highest among all treatment groups ($P < 0.05$). In CK, the relative abundance of *Lactobacillus* was only 0.97%, the lowest among all treatment groups. Zhao et al. also found that if no additives are used in high-moisture alfalfa silage, the number of lactic acid bacteria will not change significantly (36). *Pantoea*, *Rhabdanaerobium*, and *unclassified_f__Lachnospiraceae* were the main bacterial

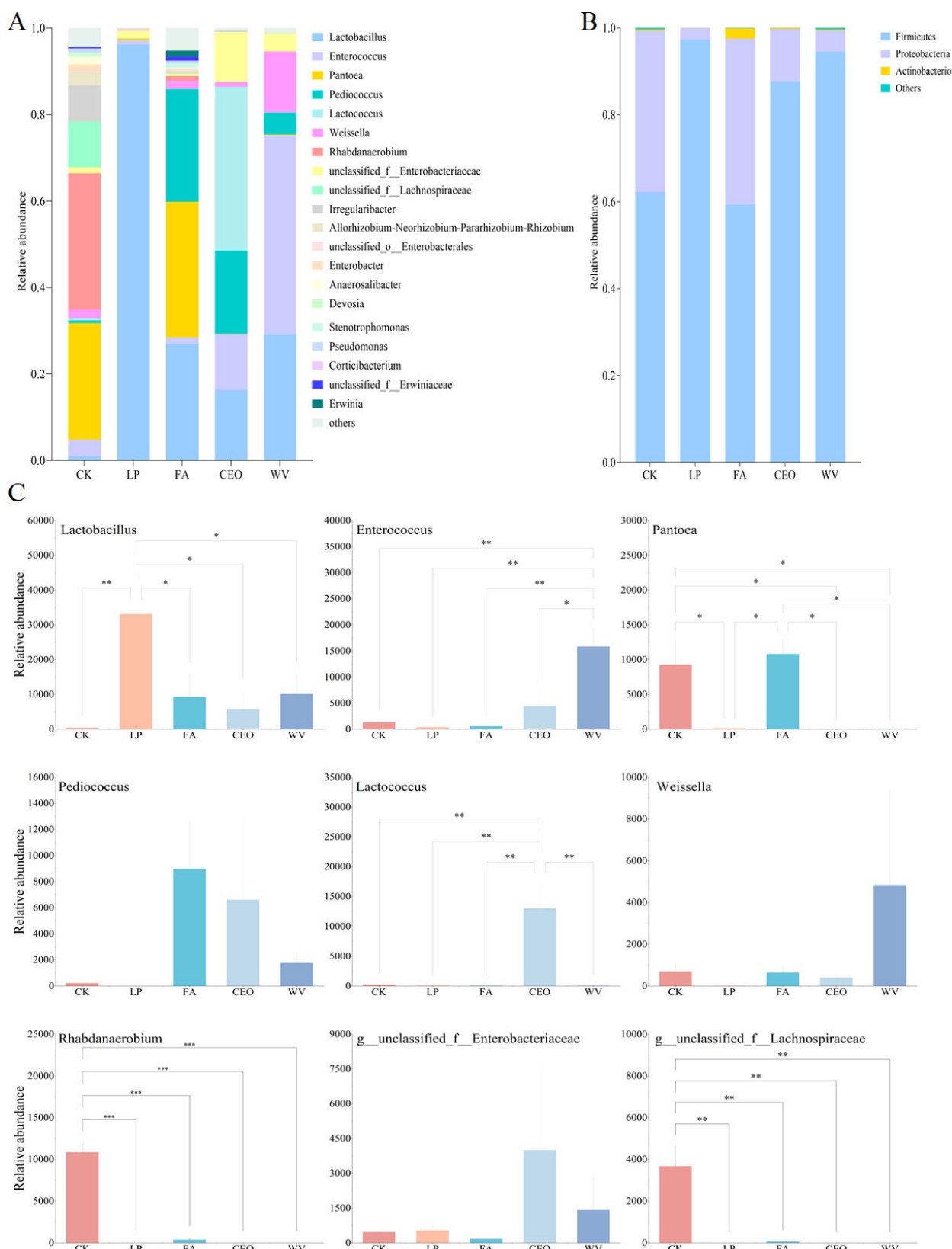

**FIG 3** Bacterial community composition at the phylum (A) and genus (B) level and relative abundance of the top nine genera in different treatment groups. One-way ANOVA of relative abundance of representative genus (C). * $P < 0.05$, ** $P < 0.01$, and *** $P < 0.001$.

communities in CK, with their relative abundances significantly higher in CK than in LP, CEO, and WV ($P < 0.05$). *Rhabdanaerobium* is an obligate anaerobe isolated from hot springs, suitable for growth in a neutral pH environment (37); hence, this bacterium was not detected in any treatment groups other than CK in the present study. *Pantoea* is thought to compete with lactic acid bacteria for fermentation substrates, thus being detrimental to silage fermentation (38). Similar to CK, the relative abundance of *Pantoea* in FA was also relatively high, at 31.37%. The abundant growth and reproduction of *Pantoea* in alfalfa silage inhibited the fermentation process of LAB, which may also be the reason for the higher ADF and NDF content in the CK and FA treatment groups. Additionally, *Lactobacillus* and *Pediococcus* were also two bacteria with relatively high abundance in FA. *Lactococcus*, *Pediococcus*, *Enterococcus*, and *Lactobacillus unclassified_f__Enterobacteriaceae* were the main bacterial species in CEO, with *Lactococcus* having the highest relative abundance at 37.90% among all treatment groups. In WV, the relative abundances of *Enterococcus* and *Weissella* were the highest among all treatment groups. *Pediococcus* is a homofermentative lactic acid bacterium that can grow rapidly in high pH environments and produce lactic acid (39). *Lactococcus* and *Weissella* can metabolize to produce lactic acid and can dominate in the initial stages of silage fermentation, ensuring the fermentation quality of alfalfa silage (40, 41). The significant presence of *Pediococcus*, *Lactococcus*, and *Weissella* is also the main reason why the FA, CEO, and WV treatment groups have much lower Lactobacillus counts than the LP group but still ultimately ferment to produce higher levels of LA.

## Bacterial co-occurrence network analysis

To investigate the influence of various additives on the interrelations among bacterial communities, a co-occurrence network of bacteria was established for different treatment groups using Pearson's rank correlation coefficient (Fig. 4). In the bacterial co-occurrence network, the nodes are ranked by their count as CK > FA > WV > CEO > LP. Meanwhile, the edges are ranked in terms of their count as WV > CK > FA > CEO > LP. The networks of all treatment groups are predominantly characterized by positive edges. Nonetheless, there is considerable variation in the proportion of positive edges among the different treatment groups. The percentage of positive edges for LP is the lowest at 58.33%, while WV has the highest at 99.21%. Banerjee et al. proposed that negative correlations in microbial co-occurrence networks indicate a potential competitive relationship for resources, while positive correlations indicate a mutually beneficial and collaborative relationship between microbes (42). This indicates that the relationship between the bacterial taxa in the treatments in this study was primarily characterized by a mutually beneficial relationship, which was particularly evident in the WV treatment, where the addition of WV enhanced the cooperative interactions between the bacteria. Zhao et al. reached the conclusion that the microbial taxa in silage with superior fermentation quality were primarily cooperative (34). The findings of this study reveal that although the fermentation quality of the alfalfa silage in the CK is relatively poor, the proportion of positive edges within the CK group is not the lowest, which differs from the results reported by Zhao et al. The discrepancy may be due to differences in the silage fermentation substrate, which could result in changes to the interrelationships within the bacterial communities. Zhang et al. demonstrated that the reduction in bacterial α-diversity resulting from high fermentation quality was the primary factor contributing to the simple structure of bacterial networks in silage, findings that are consistent with those of this study (39). In the CK treatment group, *unclassified_p__Firmicutes*, *Irregularibacter*, and *Brevundimonas* are considered key taxa within the bacterial network. In the FA group, *Cellulomonas*, Bacillus, and *Paenibacillus* are identified as pivotal members. For the LP group, *Weissella*, Lactobacillus, *Pantoea*, and *unclassified_f__Enterobacteriaceae* are recognized as significant components. In the CEO group, *Klenka*, *Sanguibacter*, *Aerococcus*, *Lactobacillus*, *unclassified_f__Lactobacillaceae*, *Enterobacter*, *Escherichia-Shigella*, *Methylobacterium-Methylorubrum*, *unclassified_f__Erwiniaceae*, and *unclassified_o__Enterobacterales* are deemed crucial. Lastly, in the WV

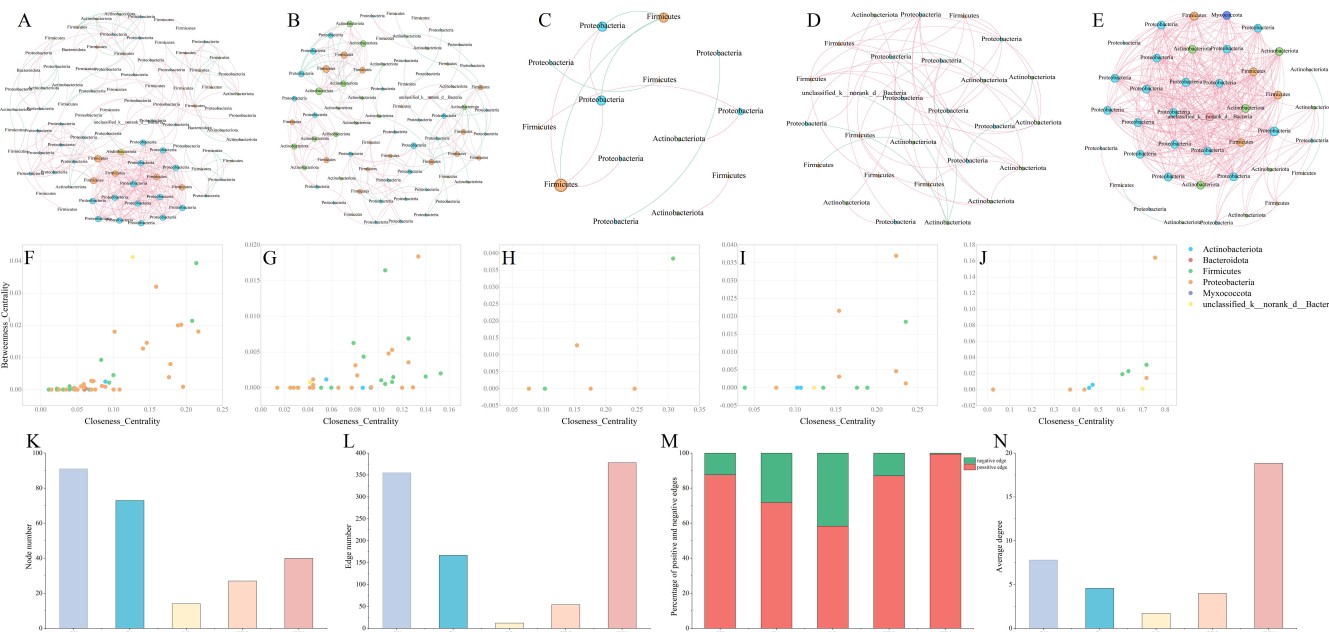

**FIG 4** Co-occurrence network of bacteria in alfalfa silage under CK (A), FA (B), LP (C), CEO (D), and WV (E) treatments. Node colors indicate bacteria at the gate level, node sizes indicate their relative abundance, red edges indicate positive correlation, and green edges indicate negative correlation. Scatter plots of node-level topological features (near-centrality and intermediate-centrality) in the networks of CK (F), FA (G), LP (H), CEO (I), and WV (J). (K) Total number of nodes in the co-occurrence network. (L) Total number of edges in the co-occurrence network. (M) Percentage of positive and negative edges. (N) Average degree of the co-occurrence network.

group, *Allorhizobium-Neorhizobium-Pararhizobium-Rhizobium*, *Lactobacillus*, and *Devosia* are acknowledged as keystone species within their bacterial network. Similar to previous research findings, the key taxa within the co-occurrence network, while having a significant impact on the bacterial community and its functions, are not necessarily the most abundant species (43). The average degree of the network was calculated, resulting in WV having the highest average degree of 18.85, followed by CK with 7.802 and LP having the lowest average degree of 1.714.

## Bacterial function and phenotype prediction

The top 20 metabolism pathways of the kyoto encyclopedia of genes and genomes (KEGG) function of bacteria in alfalfa silage under different additive treatments at level 2 are presented in Fig. 5A. Global and overview maps represent the predominant metabolic categories in each treatment group, accounting for 38.40%–40.18% of the total. Following these are categories such as carbohydrate metabolism, amino acid metabolism, and membrane transport. The LP group has the highest proportion of the global and overview maps category, followed by the CK group. The application of additives has led to an enhancement in the categories of carbohydrate metabolism, membrane transport, and lipid metabolism. Conversely, it has resulted in a decrease in the categories of amino acid metabolism, energy metabolism, and metabolism of cofactors and vitamins. The fermentation process of silage feed involves lactic acid bacteria and other microorganisms primarily utilizing the carbohydrates present in the forage as substrates, metabolizing them to produce short-chain fatty acids, predominantly lactic acid (44). This is also the reason for the relatively high abundance of carbohydrate metabolism in this study. At the same time, due to the near absence of lactic acid bacteria in the CK group, the abundance of carbohydrate metabolism is the lowest among all treatment groups. Amino acids are essential substrates for bacterial protein synthesis and primary metabolism. During silage fermentation, the amino acid metabolism of most harmful microorganisms is inhibited (34). This also explains why the

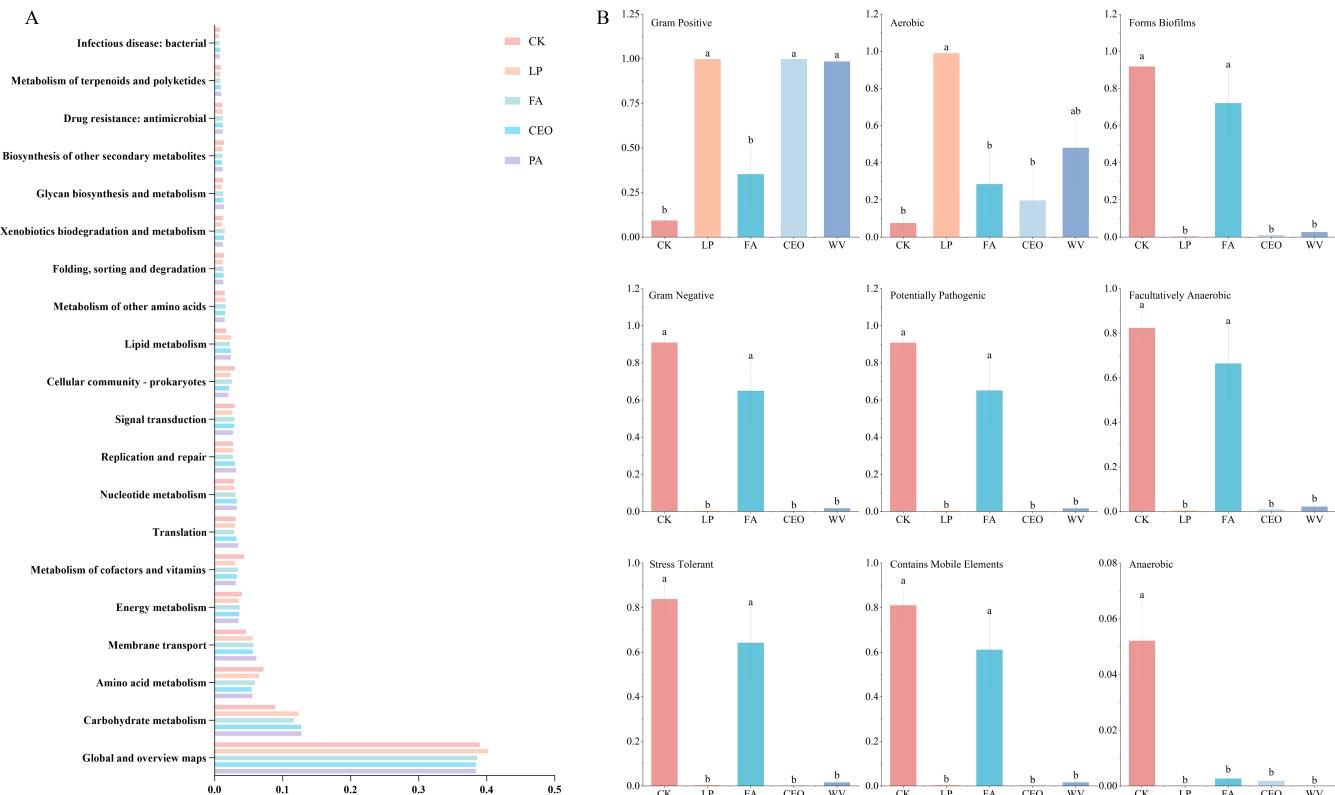

**FIG 5** The KEGG functionalities of alfalfa silage under different additive treatments primarily involved metabolism at pathway level 2 (A). Bacterial phenotypes annotation of alfalfa silage (B). Different capital letters indicate significant differences between treatments (*P* < 0.05).

level of amino acid metabolism in the CK group is higher than in the other treatment groups in this study. Previous research by Michael et al. also indicated that acidic conditions can effectively suppress amino acid metabolism induced by undesirable microorganisms, which is similar to the results of this study (45). Wang et al. found that as the fermentation process progresses, the metabolism of cofactors and vitamins in silage feed gradually decreases (46). In this study, the lactic acid bacteria fermentation in the CK (control) group was not fully effective; hence, the metabolism of cofactors and vitamins remained at a relatively high level even after 60 d of ensilage. Due to the application of additives, the energy metabolism of harmful microorganisms that are not acid-tolerant and are aerobic is inhibited. Consequently, the energy metabolism in the CK (control) group is higher compared to the other treatment groups. Simultaneously, the pattern of nucleotide metabolism changes is inverse to that of energy metabolism, which is also consistent with numerous previous research findings (46).

The microorganisms in silage feed are closely related to the immune response, growth metabolism, and diseases of livestock, and they have a significant impact on the health status and productive performance of the animals (47). The study of bacterial phenotypes in silage is important to ensure animal health and maintain performance. We therefore used BugBase to predict bacterial phenotypes in alfalfa silage (Fig. 5B). After 60 d of storage, the abundance of gram-positive and aerobic bacteria in CK was the lowest for all treatment groups. In the CK and FA treatment groups, the abundance of forms biofilm, gram-negative, potentially pathogenic, facultatively anaerobic, stress tolerant, and mobile element-containing phenotypes is significantly higher compared to the LP, CEO, and WV treatment groups (*P* < 0.05). Anaerobic abundance was also highest in CK and significantly different from the other treatment groups (*P* < 0.05). In this study, the removal effects of LP, CEO, and WV on Potentially Pathogenic and Mobile Element Containing phenotypes are very pronounced. The primary reason for this is that

harmful microorganisms such as *Staphylococcus* and *Brucella* are effectively suppressed under anaerobic and acidic conditions. Additionally, CEO can alter the permeability and integrity of bacterial cell membranes, leading to changes in the structure of membrane proteins, which in turn inhibits the growth and proliferation of pathogenic bacteria such as *E. coli* and *Staphylococcus aureus* (48). Wood vinegar contains a large amount of organic compounds such as acids, phenols, and ketones, which can also exert inhibitory effects on bacteria like *Klebsiella*, *E. coli*, and *Staphylococcus aureus* through mechanisms such as disrupting cell structures and inhibiting protein synthesis (49). Reports have indicated that gram-negative bacteria with biofilms have the potential to cause systemic infections (50). In this study, LP, CEO, and WV have shown a significant removal effect on both gram-negative bacteria and bacteria with the "forms biofilm" phenotype. This once again illustrates that LP, CEO, and WV can effectively eliminate pathogenic bacteria in alfalfa silage, which is beneficial for ensuring the health of livestock.

## Conclusions

In summary, the application of different additives significantly enhances the fermentation quality of high-moisture alfalfa silage. Post-application, the pH of the silage decreases more quickly, and it retains more DM and CP, produces more LA, and reduces the levels of ADF, NDF, AA, and BA while improving IVDMD and IVCPD. From the perspective of bacterial community composition, LP, CEO, and WV have a more pronounced inhibitory effect on harmful microorganisms such as *Pantoea* in high-moisture alfalfa silage. They also increase the number of beneficial microorganisms like *Lactobacillus*, *Pediococcus*, *Lactococcus*, and *Weissella*, promoting the lactic acid fermentation of silage feed. This leads to an improvement in the quality and safety of the alfalfa silage.

## ACKNOWLEDGMENTS

Construction of the Collaborative Innovation Base for the Key Laboratory of Forage Cultivation, Processing and High Efficient Utilization, Ministry of Agriculture and Rural Affairs.

This study was financially supported by the National Center of Pratacultural Technology Innovation (under preparation; CCPTZX2023B07).

## AUTHOR AFFILIATIONS

[1]College of Grassland Science, Inner Mongolia Agricultural University, Hohhot, China
[2]Key Laboratory of Forage Cultivation, Processing and High Efficient Utilization, Ministry of Agriculture and Rural Affairs, Inner Mongolia Agricultural University, Hohhot, China
[3]Key Laboratory of Grassland Resources, Ministry of Education, Inner Mongolia Agricultural University, Hohhot, China
[4]Inner Mongolia Academy of Agricultural and Animal Husbandry Sciences, Hohhot, China
[5]Bayannur Modern Agriculture and Animal Husbandry Development Center, Bayannur, China
[6]National Center of Pratacultural Technology Innovation (under preparation), Hohhot, China

## AUTHOR ORCIDs

Xingquan Yan ⓘ http://orcid.org/0009-0001-2731-3141
Muqier Zhao ⓘ https://orcid.org/0000-0001-8828-6151
Yushan Jia ⓘ https://orcid.org/0000-0001-7655-9933
Gentu Ge ⓘ http://orcid.org/0000-0001-9398-718X
Zhijun Wang ⓘ http://orcid.org/0000-0003-4470-7289

## FUNDING

| Funder | Grant(s) | Author(s) |
|---|---|---|
| National Center of Pratacultural Technology Innovation （under preparation） | CCPTZX2023B07 | Gentu Ge |

## AUTHOR CONTRIBUTIONS

Xingquan Yan, Conceptualization, Data curation, Formal analysis, Software, Writing – original draft | Jian Bao, Data curation, Formal analysis, Methodology | Muqier Zhao, Formal analysis, validation | Zhuotong Liu, Data curation, Software | Mingjiu Wang, Writing – review and editing | Jingyi Liu, Formal analysis | Pengbo Sun, Methodology, Software | Yushan Jia, Writing – review and editing | Gentu Ge, Conceptualization, Methodology, Resources, Writing – review and editing | Zhijun Wang, Methodology, Resources, Writing – review and editing

## DATA AVAILABILITY

The datasets generated and/or analyzed during the current study are available in the National Center for Biotechnology Information repository, PRJNA1118428.

## ADDITIONAL FILES

The following material is available online.

### Open Peer Review

**PEER REVIEW HISTORY (review-history.pdf).** An accounting of the reviewer comments and feedback.

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
