## [Reviewer comments · Microbiology Spectrum]

Microbiology Spectrum

High-Moisture Alfalfa Silage Fermentation: A Comparative Study on the Impact of Additives Including Formic Acid, *Lactobacillus plantarum*, Cinnamon Essential Oil, and Wood Vinegar

Xingquan Yan, Jian Bao, Muqier Zhao, Zhuotong Liu, Mingjiu Wang, Jingyi Liu, Pengbo Sun, Yushan Jia, Gentu Ge, and Zhijun Wang

Corresponding Author(s): Gentu Ge, Inner Mongolia Agricultural University

Review Timeline:

Submission Date:	January 3, 2025
Editorial Decision:	May 24, 2025
Revision Received:	June 4, 2025
Accepted:	June 6, 2025

Editor: Erik Hom

Reviewer(s): Disclosure of reviewer identity is with reference to reviewer comments included in decision letter(s). The following individuals involved in review of your submission have agreed to reveal their identity: Bo Yao (Reviewer #1)

Transaction Report:

DOI: <https://doi.org/10.1128/spectrum.00003-25>

Re: Spectrum00003-25 (High-Moisture Alfalfa Silage Fermentation: A Comparative Study on the Impact of Additives Including Formic Acid, Lactobacillus plantarum, Cinnamon Essential Oil, and Wood Vinegar Liquid)

Dear Prof. Gentu Ge:

Thank you for the privilege of reviewing your work. Below you will find my comments, instructions from the Spectrum editorial office, and the reviewer comments.

Please carefully address the reviewer comments and please consider including a link to the relevant sequencing data generated. in your revised manuscript.

Please also carefully revise the language/writing of the manuscript; it is clear it needs work and I suggest you consult either a native speaker of English or some language editing service to help you polish the text (<https://journals.asm.org/writing-your-paper?journal=spectrum#language-editing-services>).

Revision Guidelines

Sincerely,
Erik Hom
Editor
Microbiology Spectrum

Reviewer #1 (Comments for the Author):

This study focuses on the fermentation optimization of high-moisture alfalfa silage, systematically evaluating the effects of formic acid (FA), *Lactobacillus plantarum* (LP), cinnamon essential oil (CEO), and wood vinegar (WV) on silage quality, microbial community composition, and *in vitro* digestibility. The overall research design is well-structured, with comprehensive data supporting conclusions that hold significant reference value for the development of silage additives. Regarding data analysis, while ANOVA and Duncan's multiple comparison test are appropriate for assessing differences among multiple groups, the study does not explicitly verify whether the data meet the assumptions of normal distribution and homogeneity of variances. It is recommended to supplement the analysis with relevant statistical tests (e.g., Shapiro-Wilk test for normality and Levene's test for variance homogeneity) to ensure the robustness of the inferential results.

Reviewer #2 (Comments for the Author):

The manuscript titled "High-Moisture Alfalfa Silage Fermentation: A Comparative Study on the Impact of Additives Including Formic Acid, *Lactobacillus plantarum*, Cinnamon Essential Oil, and Wood Vinegar Liquid" explores the effects of Formic Acid (FA), *Lactobacillus plantarum* (LP), Cinnamon Essential Oil (CEO), Wood Vinegar (WV), and a control treatment (CK) on silage performance and microbial community composition. This study is expected to provide theoretical guidance and technical support for the development of novel silage additives. But I think there are some changes that need to be made, some other specific comments are listed below.

Writing needs to be improved, especially some grammatical errors, e.g., plural and singular of nouns.

"Wood Vinegar Liquid" or "Wood Vinegar" ???

Scientific names of bacterial genera should be italicized.

Consistently use "alfalfa silage" instead of "silage alfalfa."

Line 45, Supplementary Latin for alfalfa.

Line 52, "。" Should be changed to ".".

Line 93-96, The purpose of the study needs to be further elaborated.

Line 118, Consistently use "mould" throughout the text.

Line 126-127, Add details of the measurement method, e.g., information on the type of instrument used.

Line 138, Supplement the calculation formulas for IVDMD and IVCPD.

Line 170, A comma should be placed before "respectively."

Line 171-173, Use abbreviations for "the number of lactic acid bacteria"

Line 171-173, "The presence of harmful micro-organisms can cause nutrient loss from silage alfalfa and compromise silage safety. Therefore, it is necessary to promote lactic acid fermentation and inhibit the proliferation of harmful microorganisms through pathways such as the application of additives." Delete.

Line 208, "The FA treatment group had significantly higher NDF content than the other treatment groups, while the CK group had the highest ADF content ($P < 0.05$). Please explain the reason for this.

Line 247, "Iterms" should be replaced by "items".

Table 2, the units for LA, AA, PA, and BA should be changed to %FM. Keep two decimal places after the decimal point.

Line 255-256, Rewrite.

"Silage" is a noun, and the verb should be "ensiling."

Line 335, " α -diversity" is incorrectly formatted.

Line 394, P needs to be italicized.

Reference formatting needs to be checked and some documents are missing page numbers.

Reviewer #1 (Comments for the Author):

This study focuses on the fermentation optimization of high-moisture alfalfa silage, systematically evaluating the effects of formic acid (FA), *Lactobacillus plantarum* (LP), cinnamon essential oil (CEO), and wood vinegar (WV) on silage quality, microbial community composition, and in vitro digestibility. The overall research design is well-structured, with comprehensive data supporting conclusions that hold significant reference value for the development of silage additives. Regarding data analysis, while ANOVA and Duncan's multiple comparison test are appropriate for assessing differences among multiple groups, the study does not explicitly verify whether the data meet the assumptions of normal distribution and homogeneity of variances. It is recommended to supplement the analysis with relevant statistical tests (e.g., Shapiro-Wilk test for normality and Levene's test for variance homogeneity) to ensure the robustness of the inferential results.

Response: Thank you for your valuable suggestions. Based on your comments, we have conducted normality tests and homogeneity of variance tests on the relevant indicators. For indicators that do not meet the requirements for Duncan's multiple range test, the Kruskal-Wallis H test was used to assess the differences in means among treatments. The supplementary and revised data analysis can be seen at lines 158-162 of the article.

Reviewer #2 (Comments for the Author):

The manuscript titled "High-Moisture Alfalfa Silage Fermentation: A Comparative Study on the Impact of Additives Including Formic Acid, *Lactobacillus plantarum*, Cinnamon Essential Oil, and Wood Vinegar Liquid" explores the effects of Formic Acid (FA), *Lactobacillus plantarum* (LP), Cinnamon Essential Oil (CEO), Wood Vinegar (WV), and a control treatment (CK) on silage performance and microbial community composition. This study is expected to provide theoretical guidance and technical support for the development of novel silage additives. But I think there are some changes that need to be made, some other specific comments are listed below.

Writing needs to be improved, especially some grammatical errors, e.g., plural and singular of nouns.

"Wood Vinegar Liquid" or " Wood Vinegar" ???

Response: Thank you for your careful review. We have revised all the relevant descriptions in the text to "Wood Vinegar" .

Scientific names of bacterial genera should be italicized.

Response: We have changed the names of all bacterial genera in the article to italics.

Consistently use "alfalfa silage" instead of "silage alfalfa."

Response: We have revised all the relevant descriptions to "alfalfa silage".

Line 45, Supplementary Latin for alfalfa.

Response: The Latin name of alfalfa (*Medicago sativa* L.) has been added. The revised content can be seen at line 44 of the article.

Line 52, "。" Should be changed to ".".

Response: We have corrected this error. The revised content can be seen at line 51 of the article.

Line 92-96, The purpose of the study needs to be further elaborated.

Response: We have further revised the research objectives, and the results of the revision are as follows. Based on this, the present study aims to clarify the effects of cinnamon essential oil, wood vinegar, formic acid, and *Lactobacillus plantarum* on the fermentation quality and bacterial community of alfalfa silage. The focus is on whether cinnamon essential oil and wood vinegar, as novel additives, can improve the quality of silage, providing technical support for the development of new types of silage additives.

Line 118, Consistently use "mould" throughout the text.

Response: We have revised all the relevant expressions in the text to "mould".

Line 126-127, Add details of the measurement method, e.g., information on the type of instrument used.

Response: We have added the model and brand information of the Automatic Kjeldahl Nitrogen Analyzer. The revised content can be seen at line 127 of the article.

Line 138, Supplement the calculation formulas for IVDMD and IVCPD.

Response: The calculation formulas for IVDMD and IVCPD have been added to the text. They can be seen at lines 139-144 of the article.

Line 170, A comma should be placed before "respectively."

Response: The error has been corrected.

Line 171-173, Use abbreviations for "the number of lactic acid bacteria"

Response: The relevant part has been changed to an abbreviation. The revised content can be seen

at line 180 of the article.

Line 171-173, "The presence of harmful micro-organisms can cause nutrient loss from silage alfalfa and compromise silage safety. Therefore, it is necessary to promote lactic acid fermentation and inhibit the proliferation of harmful microorganisms through pathways such as the application of additives." Delete.

Response: The relevant part has been deleted.

Line 208, "The FA treatment group had significantly higher NDF content than the other treatment groups, while the CK group had the highest ADF content ($P < 0.05$)". Please explain the reason for this.

Response: Based on the analysis results of the bacterial community in this study, it is believed that the growth and reproduction of *Pantoea* may be the main reason for the relatively high contents of ADF and NDF in the CK and FA treatment groups. Since *Pantoea* competes with *lactic acid bacteria* for fermentation substrates and inhibits the fermentation process of *lactic acid bacteria*, the contents of ADF and NDF in alfalfa silage are relatively high. This part of the content has been added at lines 293-297 of the article.

Line 247, "Iterms" should be replaced by "items".

Response: We have corrected "Iterms" to "items" in Table 2.

Table 2, the units for LA, AA, PA, and BA should be changed to %FM. Keep two decimal places after the decimal point.

Response: We have changed the units for LA, AA, PA, and BA in Table 2 to % FM and retained all data to two decimal places.

Line 255-256, Rewrite.

Response: We have modified the section as follows.

To explore the effects of different additive treatments on the bacterial community in alfalfa silage, we tested the bacterial community in alfalfa silage after 60 days of fermentation.

"Silage" is a noun, and the verb should be "ensiling."

Response: We have revised the content in the article that should use "ensiling".

Line 335, " α -diversity" is incorrectly formatted.

Response: Thank you for your careful review. We have corrected this error. The revised content can be seen at line 335 of the article.

Line 394, P needs to be italicized.

Response: We have corrected that mistake. The revised content can be seen at line 395 of the article.

Reference formatting needs to be checked and some documents are missing page numbers.

Response: We have added the missing page numbers in the references.

Re: Spectrum00003-25R1 (High-Moisture Alfalfa Silage Fermentation: A Comparative Study on the Impact of Additives Including Formic Acid, Lactobacillus plantarum, Cinnamon Essential Oil, and Wood Vinegar)

Dear Prof. Gentu Ge:

Your manuscript has been accepted, and I am forwarding it to the ASM production staff for publication. Your paper will first be checked to make sure all elements meet the technical requirements. ASM staff will contact you if anything needs to be revised before copyediting and production can begin. Otherwise, you will be notified when your proofs are ready to be viewed.

Sincerely,
Erik Hom
Editor
Microbiology Spectrum